# Frailty as the Future Core Business of Public Health: Report of the Activities of the A3 Action Group of the European Innovation Partnership on Active and Healthy Ageing (EIP on AHA)

**DOI:** 10.3390/ijerph15122843

**Published:** 2018-12-13

**Authors:** Giuseppe Liotta, Silvia Ussai, Maddalena Illario, Rónán O’Caoimh, Antonio Cano, Carol Holland, Regina Roller-Winsberger, Alessandra Capanna, Chiara Grecuccio, Mariacarmela Ferraro, Francesca Paradiso, Cristina Ambrosone, Luca Morucci, Paola Scarcella, Vincenzo De Luca, Leonardo Palombi

**Affiliations:** 1Department of Biomedicine and Prevention, University of Rome “Tor Vergata”, 00133 Rome, Italy; paola.scarcella@uniroma2.it (P.S.); leonardo.palombi@gmail.com (L.P.); 2International Healthcare Programs, Lombardy Region/LISPA, 20124 Milan, Italy; ussai.silvia@gmail.com; 3Unità Operativa Dipartimentale 14 Promozione e Potenziamento dei Programmi di Healths Innovation, Direzione Generale per la Tutela della Salute ed il Coordinamento del Sistema Sanitario Regionale, Regione Campania, 80143 Naples, Italy; maddalena.illario@regione.campania.it; 4Dipartimento di Scienze Mediche Traslazionali, Università degli Studi di Napoli Federico II, 80138 Naples, Italy; 5Clinical Sciences Institute, National University of Ireland, Galway, Galway City, H91 TK33 Ireland; rocaoimh@hotmail.com; 6Department of Pediatrics, Obstetrics and Gynecology, University of Valencia-INCLIVA, 46010 Valencia, Spain; antonio.cano@uv.es; 7Centre for Ageing Research, University of Lancaster, Lancaster, LA1 4YG, UK; c.a.holland@lancaster.ac.uk; 8Department of Internal Medicine, Medical University of Graz, 8036 Graz, Austria; regina.roller-wirnsberger@medunigraz.at; 9School of Specialization in Hygiene and Medicine Prevention, University of Rome “Tor Vergata”, 00133 Rome, Italy; capannalessandra@gmail.com (A.C.); chiara.grecuccio@gmail.com (C.G.); ferraro.mariacarmela@hotmail.it (M.F.); francescaparad@gmail.com (F.P.); cristinambrosone@gmail.com (C.A.); lucamorucci88@gmail.com (L.M.); 10Unità Operativa Semplice Ricerca e Sviluppo, Azienda Ospedaliera Universitaria Federico II, 80138 Naples, Italy; vinc.deluca@gmail.com

**Keywords:** frailty, public health, community care, older adults, healthcare planning, narrative review

## Abstract

*Background:* The prevalence of frailty at population-level is expected to increase in Europe, changing the focus of Public Health. Here, we report on the activities of the A3 Action Group, focusing on managing frailty and supporting healthy ageing at community level. *Methods:* A three-phased search strategy was used to select papers published between January 2016 and May 2018. In the third phase, the first manuscript draft was sent to all A3-Action Group members who were invited to suggest additional contributions to be included in the narrative review process. *Results:* A total of 56 papers were included in this report. The A3 Action Group developed three multidimensional tools predicting short–medium term adverse outcomes. Multiple factors were highlighted by the group as useful for healthcare planning: malnutrition, polypharmacy, impairment of physical function and social isolation were targeted to mitigate frailty and its consequences. Studies focused on the management of frailty highlighted that tailored interventions can improve physical performance and reduce adverse outcomes. *Conclusions:* This review shows the importance of taking a multifaceted approach when addressing frailty at community level. From a Public Health perspective, it is vital to identify factors that contribute to successful health and social care interventions and to the health systems sustainability.

## 1. Introduction

In almost every European country, life expectancy is increasing and the proportion of people aged over 60 years is growing faster than other age groups. While undoubtedly something to be celebrated, this represents a significant challenge to public health planners and policy-makers. “Active Ageing” is an increasingly important issue in the political discussion, both at national and international levels [1]; it is described as “the process of optimizing opportunities for health, lifelong learning, participation and security to enhance quality of life (QoL) as people age” [2]. This definition reinforces the positive aspects of ageing, highlighting the importance of environmental and behavioral factors that account for variability within the ageing process as well as the role of an individuals’ “intrinsic capacity” (mental and physical capacities that change along the years) [3,4]. Most importantly, the concept of active ageing has been a driver for the implementation of an increasing number of health-promoting programs all over the world, especially in Europe [5]. However, Healthy Life Expectancy, which can be considered a proxy indicator for Active Ageing, is generally developing slower than life expectancy and in some European Union countries decreased between 2010 and 2015 [6]. Prevention seems to be proportionally less effective than clinical care and preventive strategies are poorly adopted in many countries [5,7,8].

In 2010, the European Commission launched the “European Innovation Partnership on Active and Healthy Ageing” (EIP on AHA) as a pilot project to bring together public and private stakeholders through thematic and synergic action groups with the aim of increasing by two the number of healthy life years of European citizens [1]. The EIP on AHA had three main purposes:-Improving health and QoL of older adults;-Improving the efficiency and sustainability of health systems;-Strengthening the competitiveness of European industry by investing in innovative products and services in the field of health and ageing [9].

The partnership brings together approximately 1000 stakeholders from all EU member states, and other countries outside the EU, working in six Action Groups, within a vertical structure that helps provide the EU with a multifaceted and inclusive strategy on ageing [10,11,12,13,14,15,16,17] and Reference Sites, which are quadruple helix-based ecosystems deeply interconnected with the EIP on AHA action groups [18,19]. The A3-Action Group “on Lifespan Health Promotion & Prevention of Age-Related Frailty and Diseases” proposes good practice models to achieve and support the healthy ageing of European citizens using a bottom-up approach targeting the prevention of frailty and subsequent functional decline [12]. The purpose of this paper is to report on the results of the A3 Action Group activities that specifically focus on frailty and public health in community-dwelling older adults (see Figure 1) [20].

## 2. Methods

This report adhered to guidelines outlined in the PRISMA statement. Studies were selected according to the PICOS format: Population (community-dwelling people aged more than 65 years); Intervention (Public Health interventions to mitigate frailty); Comparison (not available); Outcome (frailty and its consequences); and Study design (reviews, longitudinal studies, retrospective studies, experimental studies, impact studies).

This report reviews papers published, or submitted and accepted for publication by the A3 Action Group between January 2016 and May 2018 and written in English. The current membership of the A3 Action Group is based on two open calls to participate, launched by the European Commission on 2011 and 2015; respondents participated on a voluntary basis. [12].

A three-phased search strategy was used to select suitable papers (see Figure 2).

Studies, reports or other contributions (e.g., web page entries or blogs) were included if they were related to public health aspects of frailty and ageing, including approaches to addressing frail community-dwelling older adults in EU countries and were written by members of the A3 Action Group.

The search strategy included three phases: initially, two researchers created a preliminary draft list of published works. This list of papers was formed by the titles submitted by members and registered on the online commitment tracker tool [20]. Subsequently, a third researcher eliminated papers on the basis of the year of publication and the title (if not related to public health). Finally, six researchers excluded irrelevant papers after reviewing the abstracts. In the second phase, to maximize the number of contributions and avoid missing suitable papers, a search of PubMed, Scopus and the Google Scholar database was performed; the search used the names of selected authors suggested by the Action Group Coordinators, in addition to the following key words: “public health”, “frailty”, “comorbidity”, “quality of life”, “determinants”, “social determinants”, “community-based”, “active ageing”, “questionnaire”, “caregiver”, and “prevention”. Two researchers evaluated the results examining the title and abstract of the papers identified by this search. Finally, in the third phase, the first manuscript draft was sent to all A3-Action Group members who were invited to suggest additional contributions to be included to the manuscript.

The number of papers eligible for inclusion was 71, of which 34 were excluded for the following reasons: only focused on a specific topic or illness, such as “frailty and cancer”, “frailty and thigh-bone fracture” and “frailty and osteoarthritis” (16/34); papers addressing older populations but not specifically frailty, for example the discrimination of health status and nutritional habits (7/34); papers focused on EIP on AHA or frailty status but not both (6/34); studies conducted exclusively in countries not part of the EU (4/34); and studies on animal models, for instance on guinea pigs (1/34). The manuscript draft included 37 papers, to which, 19 papers were added in the third phase. The final report included 56 papers.

## 3. Results

The selection process produced 56 papers, divided into the following categories:(1)Frailty screening tools (8/56)(2)Assessment of frailty and main associated factors at community level (30/56)(3)Intervention protocols (13/56)(4)Impact studies (5/56)

### 3.1. Tools to Screen Frailty in Community-Dwelling Older Adults

In the A3 Group, three tools have been developed of which two validated (Table 1).

The Reference Sites Network for Prevention and Care of Frailty and Chronic Conditions in community-dwelling persons of European Union (EU) Countries (SUNFRAIL—project 664291) was a project aimed at improving the identification, prevention, and management of frailty and multi-morbidity in community-dwelling older adults in six EU countries. A specific goal of SUNFRAIL was to design an integrated model for preventing and managing frailty and multi-morbidity. It also aimed to develop the required instruments to support related clinical tasks [21,22]. In this framework a multidimensional check-list composed of 9 items (including pharmacotherapy, falls and loneliness) was developed [23]. It stands out from other tools by the absence of a rating scale component. In fact, it is based on the number of alerts associated with each item identified by the questionnaire itself, which is quite a new concept in the assessment of frailty; a first assessment of the reliability of the questionnaire has been performed comparing the alerts generated with the scores of the Loneliness scale, the Mini Mental State Examination and Walking Speed.

The Risk Instrument for Screening in the Community (RISC) is a screening and assessment risk-prediction instrument that was developed as part of the COLLaboration on AGEing (COLLAGE), to rapidly screen and stratify community-dwelling older adults attending Public Health Nursing Centres, according to their perceived risk level. This tool includes demographic data and records the presence (yes or no responses) and magnitude (mild, moderate and severe) of concern across three domains: mental state, activities of daily living (ADL) state and medical state. The instrument was designed for use by Public Health Nurses (PHNs) (i.e., community nurses). It measures one-year risk of hospitalization, institutionalization and death in community-dwelling older adults according to a five-point global risk score: from low (RISC score: 1, 2), medium (RISC score: 3) and high (RISC score: 4, 5). The RISC requires a specific enhanced training program and includes booster sessions to guarantee the correct risk assessment and the inter-rater reliability (IRR), as inadequate training might underestimate or overestimate the risk [24]. As part of the A3 action group it has been validated in community-based samples in several EU and non-EU countries [25].

The third tool is the Short Functional Geriatric Evaluation (SFGE) questionnaire that was derived from the Functional Geriatric Evaluation (FGE). The FGE is validated to predict adverse outcomes among older adults in the short–medium term. The items of the larger FGE included in the SFGE were selected by matching each FGE item with the Use of Hospital Services (UHSs), including hospital admissions, day hospital accesses and Emergency Room visits not resulting in admissions. The SFGE is not only shorter than the FGE questionnaire (smaller number of questions), but it also places a greater emphasis on socio-economic items, which are strongly related to the use of services by community-dwelling older adults. The SFGE cut-off score for frailty included the construct of pre-frailty in order to identify the entire population generating the highest demand for care services [26].

Frailty screening and assessment is a fundamental issue in public health to allow for the planning of prevention programs and preventative services. Staff can be trained easily how to score these tools, all of which can be administered at community level, in primary social or health care settings. The debate around the appropriateness of tools for frailty screening is still ongoing [27,28] with insufficient evidence for screening, monitoring or surveillance programs at population level in Europe. All the tools developed by A3 Action Group members considered the biopsychosocial dimension of frailty, which is particularly important at community level. They were all useful in predicting negative outcomes in the short–medium term. An umbrella review conducted by A3 Action Group researchers of tools to screen for frailty, published before the validation of these new instruments, did not find any short multidimensional screening tools suitable for use by public health practitioners at population level [29]. The proposed tools represent a crucial contribute towards the feasibility of frailty screening in large population at community level. Given this, the tools developed as part of the EIP on AHA represent a crucial contribution towards developing valid frailty screening instruments for epidemiological studies.

### 3.2. Assessment of Frailty and Associated Predictors at Community Level

The main goal of this section is to identify epidemiological factors useful in planning optimal social and health care services and interventions for frail older people. The main characteristics of the studies published by A3 Action Group members are summarized in Table 2.

#### 3.2.1. The Prevalence and Predictors of Frailty

In a sample of Italians aged over-64 years and representative of a regional geographic area which included the capital city of Rome, the prevalence of frailty was 21.5%; it increased to 31.8% when only those aged over-74 were considered [28]. Many variables were closely associated with frailty such as the level of disability and presence and type of those cohabitants. “Living with a spouse” and “having a high educational level” were important protective factors against developing frailty [30].

Frailty was also associated with higher use of hospital services and mortality; the group showing the highest rate of use was a cluster made up of pre-frail and frail individuals [29]. The major determinants of rates of UHSs were the functional, the psychological and the socio-economic domains; the presence of specific cardiac and renal diseases were associated with a higher use of hospital services only among the frail/very frail [31]. Mortality was related to a low socioeconomic status and to a low level of psychosocial supports [32].

A better understanding of the main determinants of care was provided in a Portuguese study showing that the presence and the magnitude of mental health concerns scored on the RISC were associated with higher rates of institutionalization from the community [33]. Participants in nursing homes had a higher proportion of mental health conditions and cognitive impairment, in particular severe mental health concerns. After exploring the relationship between the type of care services (nursing homes, day centres and home care services) and the perceived risk of adverse outcomes (institutionalization, hospitalization and death), the research group observed that the highest risk of hospitalization and death was among nursing home residents: those in nursing homes had a higher risk of hospitalization and death (84.3% and 80.9% respectively) than persons receiving home care services (64.3% and 54.4% respectively) and attending day centers (64.4% and 57.8% respectively). Those attending day care centres had a greater risk of institutionalization than home care service users [33]. The risk of adverse events in nursing home residents depended on correct allocation and type of services offered. These should receive an evaluation based on social criteria, but also an assessment of their physical and mental requirements to ensure the allocation of adequate assistance from appropriate staff. The use of a pre-screening tool could, therefore, not only identify the most appropriate setting it could improve allocation of limited resources, potentially reducing the occurrence of negative health outcomes [33].

#### 3.2.2. Frailty and Multimorbidity

The evidence found supports the notions that older adults affected by multimorbidity have lower QoL, a higher number of medical visits, a greater risk of impairments in basic and instrumental ADL and more hospital admissions [34]. For these reasons, multi-morbidity is identified as a major determinant of high health care service use and is associated with the development of frailty [30,33,34,35]. However, the stability of disease and its impact on socio-functional factors is a better predictor of the risk of death than the number or type of chronic diseases [31,34,36]. Overall, these results show the relevance and impact of an individuals’ functional and socio-economic status on their use of health services, i.e., these are more important than the type of disease. Thus, preventive actions taken by public health institutions to manage multi-morbidity that focus on broader psychosocial and functional factors can impact on frailty and improve the QoL of older community-dwellers [37].

#### 3.2.3. The Caregiver Network

The growing role played by caregivers in addressing frailty at population-level is evident from a large meta-synthesis recently published by A3 researchers as part of the EU-funded Frailty Management Optimisation through EIP-AHA Commitments and Utilisation of Stakeholders Input (FOCUS—project 664367) study.

According to this, the provision of care must involve the entire caregiver network, both formal and informal, to identify needs and best coping strategies [38]. The results of a Portuguese study showed a strong association between the risk of death and the ability of carers to manage older adults’ healthcare requirements: the perceived risk of death was 65 times higher in cases where the caregiver was unable to manage either mental concerns, physical concerns and/or ADL impairments [36]. Furthermore, an increased risk of elder abuse was observed in the informal caregiver network in case of the highest complexity of the care needs [39].

The presence of a spouse was “protective” against institutionalization within one-year of assessment, while the use of formal caregiver resources (state or privately funded) ensured better management of those with complex care needs [40].

The perception of policy-makers was also highlighted by a qualitative study, which shows the need for integrated public health approaches to manage frailty [41]. The role of training seems to be crucial in supporting informal caregivers; it should be a main focus for formal health services in order to support society in taking care of older relatives/friends [41].

#### 3.2.4. Frailty and Quality of Life

Multiple associations have been identified between psychological and mental discomfort and social factors contributing to higher levels of frailty and lower QoL by A3 researchers. Impairment in cognitive function is strongly related to frailty and is associated with impaired awareness of disease (insight), depression and disinhibition [42] while cognitive reserve (the ability of the brain to develop alternate cognitive strategies to solve a problem despite impaired cognition) is related to a better QoL [43,44,45]. Furthermore, anxiety and depression were associated with physical disability, lower QoL and a high health service uses (i.e., hospitalization in the last year, number of medical visits) [46].

Males seem to benefit from social support, while women benefit most from social and cultural participation [43,47,48]. Social support and participation are both associated to better QoL and lower levels of frailty. Lower socio-economic status is also correlated with higher mortality and is connected with poor social supports [32]. Regarding loneliness (the subjective perception of social isolation) and social isolation (an objective and measurable condition), a close relationship was highlighted between isolation and psychological well-being and resilience (ability to adapt to change) [48], health status [49] and a poor social network, living alone [50]. This strong relationship between psycho-social factors, frailty and QoL was also studied using the World Health Organization Quality of Life Assessment AGE (WHOQOL-AGE) scale, used to compare the QoL of older and younger adults [51].

#### 3.2.5. Frailty and Nutrition

Good nutrition plays an important role in ensuring healthy ageing. This topic was explored in a recent study where approximately 23% of Portuguese community-dwelling older adults reported food insecurity, in particular those aged 70–74 years. The prevalence of the food insecurity was higher among women, older adults who were overweight/obese and in those with low household incomes [52] and was also associated to stopping medication and decrease in medical visits.

#### 3.2.6. Physical Ability: Mobility and Physical Activities

In an effort to identify the physio-pathological background leading to the onset and progression of physical frailty, the association between sarcopenia and potential risk factors and biomarkers have been studied [53,54]: Pre-frailty, being overweight and depression are identified as significant risk factors for frailty [53]. Being overweight is associated with the development of important alterations in muscle mass and function in robust older people that significantly contributes to subsequent frailty [54]. Based on the analysis of the underlying physio-pathological mechanisms, the same research group found that melatonin use might limit or slow sarcopenia [55] and the progression of frailty. A recent study found that a per-unit improvement in mobility was significantly associated with a 2% reduction in the risk of death, while practicing a high level of physical activity (PA) was significantly associated with a 51% lower risk of death [56]. Furthermore, an appropriate amount of PA and high mobility levels are connected to better functional capacity [57] and a lower risk of death [56], while sedentary behaviors are highly prevalent among older adults and associated with increased risk of physical dependence [58,59]. Improvement in physical performance is also related to a reduction in the risk of institutionalization [60].

In conclusion, based on the results of the selected studies, frailty and its major determinants are related to increased health services use and health costs as well as a reduction in QoL and negative health outcomes. At the community level factors leading to the development or worsening of frailty are multidimensional and include psychophysical, environmental and social components. In order to prevent the onset of frailty, Public Health should invest in interventions or programs aimed at improving the management of co-morbidities with a focus on cognitive and functional impairment, supporting the caregiver’s ability to manage frail older persons, preventing social isolation and malnutrition, and promoting PA.

### 3.3. Interventions to Mitigate Frailty and Its Consequences

Several intervention programs were implemented between 2016 and 2017 in the context of the framework of the EIP-AHA A3 Action Group activities (Table 3), many of which are funded by the EU.

The FOCUS project aims to contribute to the reduction of the burden created by frailty, by designing evidence-based protocols in order to introduce innovative practices in the management of frailty. The partners contributing to this multi-centre project are working on advancing knowledge of frailty detection, assessment, and management, including biological, clinical, cognitive and psychosocial markers, in order to change the paradigm of frailty care from acute intervention to prevention and rehabilitation. The outcomes include the generation of new guidelines and recommendations to increase success of interventions and the creation of a technological platform for EIP-AHA partners and stakeholders [45,61,62]. The guidelines and recommendations were piloted in five sites (Valencia in Spain, Milan in Italy, Coimbra in Portugal, Wroclaw in Poland and Birmingham in the UK).

The FrailSafe project (project 690140) aims to use technology and clinical research to better understand frailty, quantify it and eventually prevent it by analyzing physiological, cognitive, behavioral and social parameters in real-time. Through the developed IT dashboard, personalized interventions will be communicated to the older person and enable him/her to adopt a pro-active behavior by monitoring his/her health. Currently, the project is being tested in three pilot sites in Greece, France and Cyprus. The validation phases will take place at the end of 2018 and will provide results regarding the effectiveness of this innovative system to better prevent frailty [63].

The Prevention of Malnutrition In Senior Subject (PROMISS—project 678732) aims to better understand and ultimately prevent protein energy malnutrition in community dwelling older adults in Europe [64]. The PROMISS project developed and validated a short food questionnaire called the Protein Screener 55+ (Pro^55+^) to screen for low protein intake in community-dwelling older people [65]. The Pro^55+^ performed well in discrimination of community-dwelling older persons with a low and high protein intake [64,65].

Other intervention projects are focused specifically on physical activity, nutrition and polypharmacy. A Portuguese working group developed a multi-component community-based exercise intervention to improve the gait pattern and functional fitness of a group of community-dwelling older people. They hypothesized that a community-based program will have significantly better results in the improvement of gait and functional fitness parameters if it is regularly repeated. This intervention is based on a baseline assessment carried out using validated physical tests [66]. Similarly, the recently started Frailty, Falls and Functional loss Education (3Fights@edu) program is a massive open online course directed specifically for older people living in the community and their families which is intended to (1) give a comprehensive perspective about frailty, falls and functional decline with aging, and (2) provide strategies to promote active ageing and maintain independent living [67].

The H2020 i-PROGNOSIS project (project 690494) includes a Personalized Game Suit (PGS) that aims to mitigate the frailty symptoms in a personalized and gamified environment, involving Serious Games (SGs). The PGS design introduces, in a unified platform, the integration of different SGs, i.e., ExerGames, DietaryGames, EmoGames, and Handwriting/Voice Games, all related to Parkinson’s Disease (PD) symptoms. From this perspective, in the i-PROGNOSIS PGS platform, various modules within a holistic technological environment are under development towards older adult’s physical and emotional status monitoring and support [68].

A nutritional approach called NutriLive also aims at preventing functional decline and frailty across the EU, by implementing a structured Screening Assessment & Monitoring-Pyramid-Model (SAM-AP) into various health and social care systems. This pyramid model shows the ranking variables or tools needed to screen, assess, monitor, prevent or treat malnutrition in consultation with health care professionals. This approach could be integrated into the education of healthcare professionals such as medical doctors, dieticians, nurses and nurses’ aides. This suggests that there is a need to establish a new discipline of chefs, better educated to address the nutritional needs of older adults [69].

A working group mainly based at the University Federico II—Naples suggests a computerized prescription support system combined with CGA in order to manage polypharmacy among older adults. Preliminary data demonstrated this combination improves drug use and has positive effects on health outcomes, including a reduction in adverse drug reactions [70].

In the Personalised ICT Supported Service for Independent Living and Active Ageing (PERSSILAA—FP7-ICT-610359) project, technology supported self-management programs for PA, cognition, and nutrition were developed and offered in a gamified environment. Results show that older adults are able to use this environment and that when used long term, adherence to exercise programs is very high. Notably, the benefits of adequate nutrition are also evident from the preliminary data of the program [71,72].

The Lazio Regional heat prevention plan started in 2006 and specifically addressed frail older subgroups. The program, aimed at protecting older adults from the heat waves, is based on the active surveillance of high and very high-risk patients by General Practitioners (GPs), who received the list of frail patients by the regional Department of Epidemiology. The list is compiled on the basis of clinical and administrative information gathered on a routine basis by the Regional Health Service. Every year, an average of 550 GPs participates on a voluntary basis with the program and around 19,000 patients are surveyed each summer. This active surveillance by GPs is based on monitoring patient health status, adjusting pharmacological treatments, and adhering to emergency protocols in collaboration with local health services, hospitals and nursing homes [73].

All these protocols developed in the framework of the A3 Action Group highlight the potential impact of intervention programs based on tackling or identifying key components of frailty at population-level. These contribute to the development of a person-tailored approach to the management of frailty at community level in order to prevent specific factors contributing to frailty including physical decline, malnutrition and adverse drug reactions.

### 3.4. Impact Studies

The main characteristics of studies, included in this section, are summarized in Table 4.

A randomized controlled trial in older adults aged >65 years, confirmed that PA is important to prevent frailty and showed the effectiveness of a multi-factorial intervention program to modify physical parameters, neuro-cognitive parameters and medication [74]. The intervention group received exercise training, intake of hyperproteic nutritional shakes, memory training, and medication review for 12 weeks.

This study highlighted that after 3 and 18 months, in the intervention group compared with the control one:-Short physical performance Battery (SPPB) score improved to 1.58 to 1.36 (*p* < 0.001);-handgrip strength increased to 2.84 and 2.49 kilogram (*p* < 0.001);-functional reach rose to 4.3 and 4.52 centimetres (*p* < 0.001);-the number of prescriptions decreased to 1.39 and 1.09 (*p* < 0.001);-there was an improvement in neurocognitive battery scores [73].

Behm et al. analyzed the impact of a preventive home visit or multi-professional senior group meetings on the progression of frailty and found that the intervention was able to slow the progression of frailty but not to prevent the onset of frailty in community-dwellers aged over-80 [75].

Another program aimed at increasing social capital in community-dwelling older adults, the “Long Live the Elderly!” (LLE) program in Rome, Italy was able to limit the limit the increase of mortality during the 2015 heat wave by approximately 50% with a reduction of the expected mortality by 13% [76]. Moreover, the LLE program seemed to reduce the acute hospital admission rate by approximately 10% in a sample of older adults during the first six months of follow up [77]. The LLE program is based on a pro-active approach targeting all the over-75 residents in the operational area with a special focus on the over-80 because of their increased susceptibility to acute event especially during extreme climate events. The first step is the administration of a short questionnaire to screen frail individuals who then receive a personalized intervention plan. LLE aims at counteracting social isolation and acts as a one-stop shop to access all the other health and social services. The program uses phone calls to pro-actively monitor clients throughout the year according to their level of frailty. During extreme weather (heat waves/cold spells) all the participants receive periodical phone calls to receive advice and practical help to protect them from the dangerous consequences of extreme climatic conditions. Where required home visits and specific interventions (for example bringing food and medicine during an episode of illness or providing assistance with complex instrumental activities including navigating healthcare systems) can be provided by the program’s operators or by volunteers.

A detailed review of the impact of frailty management intervention based on data from papers published before 2016 by the A3 Action Group, showed mixed results overall, albeit they highlighted the need for more investigation of individuals with different level of frailty and how these can be managed at population-level [62]. Physical exercise programs were shown to be generally effective for reducing or delaying the onset of frailty but only when conducted in groups. Favorable effects on frailty indicators were also observed using nutritional supplementation, cognitive training and combined multi-component interventions e.g., combining physical exercise with nutritional supplementation [61]. While more research is needed to confirm these findings and their utility in the long-term, the potential benefits of such multi-intervention programs as preventative strategies are encouraging and suggests that they should be prioritized for further evaluation [61,75].

## 4. Discussion

This paper presents activities of the A3 Action Group of the EIP on AHA delivered between 2016 and 2018, focusing on public health research related to frailty. It shows the importance of taking a multifaceted approach when addressing frailty in community-dwelling older adults. This report highlights the depth and breadth of research conducted by the A3 group. From a public health perspective, frailty is a relevant construct to target because it allows planners to stratify populations according to their risk of negative outcomes [78]. The use of short questionnaires, like the ones developed by the A3 working group, allow for the efficient risk-stratification of communities and given evidence that few instruments are suitable (sufficiently accurate and reliable with short administration times) for use in clinical practice [79], those developed as commitments under the EIP on AHA have potential. They may also serve to support implementation of frailty screening programs at population level (at least the part of the population at higher risk of frailty including those aged over 75). This represents a fundamental step in managing frailty among community-dwelling older adults and is particularly important given results from the ongoing EU-funded Joint Action on Frailty Prevention, ADVANTAGE (3rd Health Programme grant number #724099), that there is currently insufficient evidence for frailty screening, monitoring and surveillance strategies to manage frailty in Europe [28]. Further, although frailty is now recognized for the first time as an emerging public health emergency [80], there remains insufficient, albeit growing evidence for approaches to prevent its onset in pre-frail subjects [81]. Instruments and pathways to support this early identification are, therefore, important.

Risk identification and stratification were major themes explored by many A3 Action Group researchers. This composite risk is related to frailty, which itself is a combination of many factors including socio-economic status, psycho-physical health and environmental characteristics [82]. The concept of frailty is an effective approach that can be used to establish priorities in accessing social and health services for older adults, to plan integrated care tailored to the individual and to measure the need for care at population level in order to allocate sufficient human and financial resources to appropriate community care services [80]. Undoubtedly, this is an aspect that public health planners and policy-makers should take into consideration [80]. In addition to the risk-stratification of populations, through multidimensional prognostic indexes [79], further attention should be devoted to the identification of pre-disability, which may be more amenable to proactive preventive multi-domain interventions. The results of A3 commitments and studies produced by the Action Group help better our understanding of the risks associated with developing frailty including the factors that drive frailty transitions from non-frail and pre-frail to frailty and back, some of which are socially determined [83]. The results obtained by the Action Group highlight several elements that could be combined in an intervention model to prevent frailty and functional decline as well as to slow their progression in order to mitigate the impact of these factors on individual health. This paper shows that this approach is able to re-direct the use of health care resources and it is likely associated with better quality of life. However, a model of community care based on frailty assessment and management is still lacking [28], because of insufficient evidence for the positive impact of this approach.

The main limitation of the A3 working group is that closer collaboration among different research groups working in this field across Europe is needed to develop and trial an approach that combines different programs in a unique exploratory framework. This approach will likely promote effective synergies, able to show pool resources and identify meaningful effects. There is an urgent need to involve regulatory authorities, private partners, and non-profit organizations in order to promote the implementation of multifaceted programs at community level that can exploit the move towards integrated health and social care to better manage frailty and multimorbidity and prevent subsequent functional decline [84]. While a variety of approaches in different areas and settings in regions of the EU are likely required, good practice models, based on the management of frailty at community level and the integration of social and health care, will offer a strong conceptual framework to support and develop locally-based responses. Another limitation of this paper is the potential bias resulting from an analysis based only on the activities of the A3 working group members. Further, as with any review it is possible some studies were not included and the review may be susceptible to publication bias. That said, the three-phase approach to identifying suitable material and the attempt made to contact all A3 Action Group members means that these were likely minimized.

From a research perspective, frailty is an essential paradigm to allow meaningful comparisons of older adults with increasingly complex socio-economic environments at risk of adverse health outcomes. It is important to understand factors contributing to successful social and health interventions to address frailty. These will become an important part of public health strategies designed to address our ageing demographics. The impact of new instruments and interventions focused on the management of frailty at community level presented in this review is potentially great, albeit more study is now required to confirm these findings and to better understand the effect and interplay between factors that can promote successful ageing including physical activity and nutrition. Robust methodological approaches are required to measure the impact of interventions; this is crucial in order to guide the future development of community care and to contribute to the medium-long term sustainability of health care systems in the face of population ageing in the EU and beyond. This report shows the strategic role that the A3 Action Group of the EIP on AHA has had since its inception and its potential to expand the public health agenda related to ageing in Europe. This can be achieved by supporting not only evidence-based approaches to address frailty, but also the development of policy and guidance documents. More effort is required to create an effective platform between public health and clinical practice.

## 5. Conclusions

This report highlights how the A3 Action Group of the EIP on AHA has developed approaches to identify and address an increasingly important aspect of public health: the ageing of populations and the high and growing prevalence of frailty and functional decline in Europe, particularly at population level [85]. A number of useful screening and risk-stratification instruments were developed by the Action Group and the multidimensional interdisciplinary approach taken has helped identify and reinforce the importance of factors that play an important role in not only the development of frailty but also how to achieve optimal outcomes using health and social care interventions. More impact studies, with well-constructed methodologies, must be conducted to further evaluate the factors and possible interventions identified by the A3 Action Group and summarized in this report. The development of an effective model for health promotion and care addressing the needs of community-dwelling older adults is required and the feasibility, efficiency and effectiveness of managing frailty in the community should be tested under ‘real world’ conditions.

## Figures and Tables

**Figure 1 ijerph-15-02843-f001:**
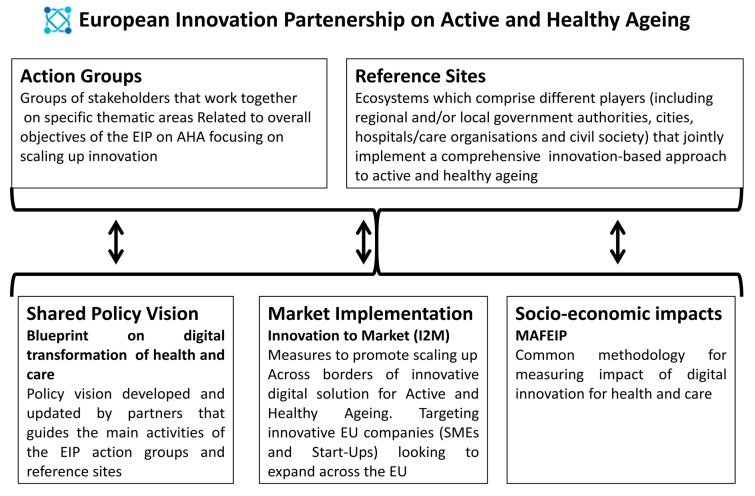
The structure of the European Innovation Partnership on Active and Healthy Ageing (EIP on AHA).

**Figure 2 ijerph-15-02843-f002:**
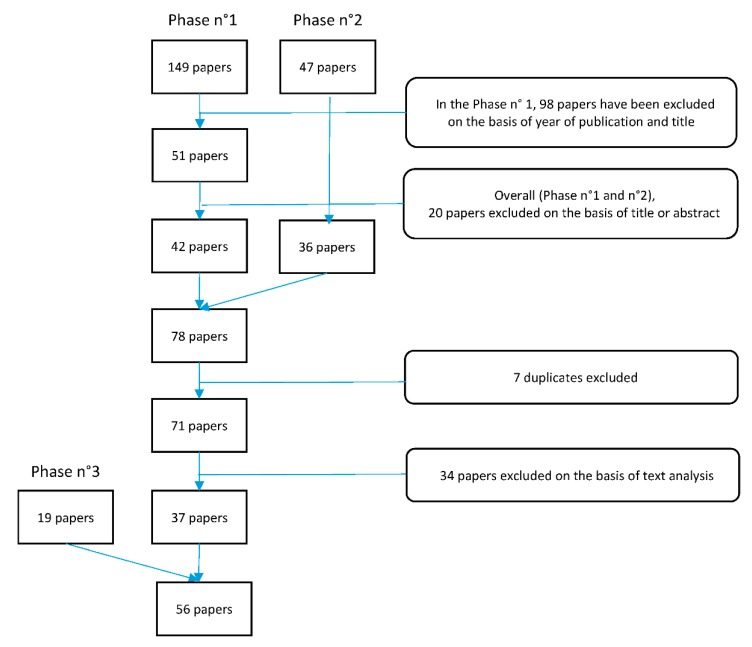
Flow chart of the search method.

**Table 1 ijerph-15-02843-t001:** Tools to assess Frailty or Quality of Life developed within the activities of the A3 group.

Article (Ref. Number)	Tool Description	Way of Administration	Developer	The Tool Has Been Tested (a), Implemented (b), or Validated (c)	Country Where It Has Been Tested/Validated
SUNFRAIL Tool [22]	9 items exploring socioeconomic domain, Psycho-Physical status, functional capacity	Social and/or Health care personnel	SUNFRAIL EU project	(a)	Italy
The Risk Instrument for Screening in the Community (RISC) [24]	32 items exploring domains of physical condition, mental health, functional status, community support, housing, social relationships	Public Health nurses	COLLAGE *	(a), (b), (c)	Ireland, Spain, Portugal, Australia
Short Functional Geriatric Evaluation (SFGE) [26]	13-items exploring socioeconomic domain, Psycho-Physical status, functional capacity	Personnel with secondary school diploma	University of Rome “Tor Vergata”	(a), (b), (c) (for predicting the Use of Hospital Services)	Italy

* COLLAGE: COLLaboration on AGEing.

**Table 2 ijerph-15-02843-t002:** Frailty and associated predictors.

First Authors and Ref. Number	Study Design	Sample Size	Sample Age	Instrument(s)	Main Outcomes	Country	Follow Up
The Prevalence and Predictors of Frailty							
Liotta [28]	Cross sectional	1331, females 54.2%	Over 64 years	Functional Geriatric Evaluation	A total of 21.5% of frail individuals, 31.8% among the over-74 years; factors associated to Frailty: more than one neurologic disease, disability, low education, living alone, being older than 84 years.	Italy	NA
Gilardi [29]	Observational longitudinal cohort study	1280, females 54.4%	Over 64 years	Functional Geriatric Evaluation	Mortality rate: 1.8%, 10.1% and 19.1% among robust, frail and very frail respectively; UHS rate was 957.4 for frail/pre-frail and 594.5 for robust. Factors associated to highest UHS rate: disability, lack of social resources, psychological/psychiatric impairment, physical impairment, lack of home care.	Italy	1 year
Domenech-Abella [30]	Longitudinal survey	2783, females 54.6%	Over 50 years	Set of instruments	Psychosocial and biomedical well-being as well as Socioeconomic Status (SES) had a role in the prediction of mortality: adults who had lower levels of psychosocial SA were more prone to die, independently of SES; significant interaction was observed between biomedical SA and SES (*p* = 0.046).	Spain	3 years
Teixeira [31]	Cross sectional	224, females 69.6%	Over 64 years	Risk Instrument for Screening in the Community	The 64.4% of the clients of home care services and day center services was at risk of hospitalization; over the 50% of the clients of home care services and day center services was at risk of death; the 73.3% of the clients of day centers was at risk of institutionalization.	Portugal	NA
Frailty and Multimorbidity (see also [28,29,31])							
Olaya [32]	Longitudinal survey	3541, females 54.5%	Over 50 years	Set of instruments	Patients with showed an increased percentage of hospital admissions 16.8% vs. 30.1% or 44.5), and medical visits in the last 12 months (3.04 vs. 5.55 or 7.02).	Spain	3 years
Rodrigues [33]	Cross sectional	2393, females 55.8%	Over 65 years	Set of instruments	Multimorbidity prevalence: 78.3% of the adults aged 65–69 years and 83.4% among the over-80 years. 25.8% of the sample was hospitalized in the last year.	Portugal	NA
Teixeira [34]	Cross sectional	4470, females 58.7%	Over 64 years	Risk Instrument for Screening in the Community	The perceived risk of death increased with the increase of severity of medical concerns (OR: 1.6 for mild severity; 9.7 for moderate severity; 48.6 for severe) and the decrease ability of caregiver to manage (OR: 4.5 for “can manage”; 65.3 for “cannot manage”).	Portugal	NA
Garin [35]	Cross-sectional	41,909, females 52.3%	Over 50 years	Questionnaire to gather information on health and well-being	Multimorbidity prevalence increases with age. The factors that show higher odds for multimorbidity are: higher age, female, lower education, separated/divorced/widowed and rural inhabitance. Multimorbidity patterns identified across countries: Cardio-respiratory, metabolic, mental-articular and respiratory pattern.	Finland, Poland, Spain, China, Ghana, India, Russia, Mexico, South Africa	NA
The Caregiver Network (see also [34])							
D’Avanzo [36]	Meta-synthesis of qualitative evidence	45 studies	Older adults, caregivers		A bottom-up approach involving formal and informal caregivers is needed to approach frailty as a malleable and preventable condition.	Western countries	NA
Orfila [37]	Cross sectional	829, females 82.8%	Caregivers	Set of instruments	Prevalence of abuse risk by the caregiver 33.4%; factors associated: caregiver burden (OR = 2.75; 95% CI: 1.74–4.33), caregiver anxiety (OR = 2.06; 95% CI: 1.40–3.02), caregiver perception of aggressive behavior in the care recipient (OR = 7.24; 95% CI: 4.99–10.51), and a bad previous relationship (OR = 4.66; 95% CI: 1.25–17.4).	Spain	NA
O’Caoimh [38]	Observational prospective cohort study	803, females 64%	Over 65 years	Risk Instrument for Screening in the Community	Risk of institutionalization is associated to the caregivers’ difficulty in managing medical issues (OR = 3.8; 2.22–6.86); the caregivers’ difficulties are not associated with higher risk of death/hospitalization.	Ireland	1 year
Gwyther [39]	Thematic analysis of semi structured qualitative interviews	7	Health care policy makers	Ad Hoc	‘Knowledge gap’, around frailty and *awareness of the malleability of frailty.* Frailty should be recognized as a clinical syndrome and managed by integrating social and health care. Need for a culture shift to overcome the silos approach in providing care.	UK, Italy, Spain, Poland, representatives of EU	NA
Frailty and Quality of Life (QoL)							
Amanzio [40]	Cross sectional	60, females 63.3%	Over 50 years	Set of instruments	Frailty is associated to action monitoring and monetary gain (cognitive domain), depression and disinhibition (behavioral domain).	Italy	NA
Raggi [41]	Cross sectional	5639, females 51.2%	Over 18 years	WHOQOL-AGE	The model explained 45% of the Quality of Life variation: The biggest variation was related to social and demographic variables (22.5%), followed by chronic condition (4.6%).	Finland, Poland and Spain	NA
Lara [42]	Cross sectional	1973, females 56%	Over 50 years	Set of instruments	Cognitive Reserve was associated with higher QoL and this association was mediated by disability, which explained about half of the association, and depression and cognition that explained 6–10% of this association.	Spain	NA
Gwyther [43]	Review				Healthcare interventions were successful when they were (1) sufficiently different from usual care; (2) based on health psychology; (3) offering choice over intervention elements; (4) organized in group settings; (5) multi-component (exercise, cognitive, nutrition, social).	NA	NA
de Sousa [44]	Cross sectional	1680, females 54%	Over 64 years	Set of instruments	The estimated prevalence of anxiety was 9.6% and depression is 11.8%. Anxiety and depression were associated to higher levels of physical disability (OR = 3.10; 96% CI: 2.12–4.52; OR = 3.08, 95% CI: 2.29–4.14) and lower levels of quality of life (OR = 0.03, 95% CI: 0.01–0.09; OR = 0.03, 95% CI: 0.01–0.06), respectively.	Portugal	NA
Tobiasz-Adamczyk [45]	Cross sectional	5099, females 58.6%	Over 50 years	WHOQOL-AGE	Males benefited more (in QoL) from social networks and social support, and women from social participation. Gender-related differences (in QoL) were associated with social networks in the group of 80+, for social support in the 50–64 and 65–79 years, and for social participation in the 65–79 years.	Finland, Poland and Spain	NA
Raparacciuolo [46]	Cross sectional	571, females 50%	Over 60 years	Set of instruments	Better Resilience and Psychological Well-Being are associated to social participation to cultural activities. Participating subjects are more likely to adhere to diet/nutritional regimen.	Italy	NA
Rico-Uribe [47]	Cross sectional	10,800, females 57.4%	Over 18 years	UCLA Loneliness Scale	Loneliness increases in over-79 population; higher age, the presence of depression and a higher score on loneliness were associated with a worse health status.	Finland, Poland and Spain	NA
Domenech-Abella [48]	Cross sectional	3535, females 45.9%	Over 50 years	Set of instruments	Feelings of loneliness or depression were reported in the 13% and 12.1% of the sample, respectively. They were associated with the size and the quality of the network as well as with the, frequency of contact. Small social network was observed among the adults with depression and feelings of loneliness.	Spain	NA
Santos [49]	Cross sectional	9987	Over 18 years	WHOQOL-AGE	Respondents from Finland, Poland, and Spain attribute the same meaning to the latent construct studied, showing the reliability of the used tool.	Finland, Poland and Spain	NA
Fernandes [50]	Cross sectional	1885, females 55.5%	Over 64 years	Set of instruments	A total of 23% of older adult reported to be food insecure; factors associated with food insecurity were gender (to be female) older age, financial difficulties lower education, living in the Azores and Madeira, stopping medication and medical visits, higher multimorbidity.	Portugal	3 years
Physical Ability: Mobility and Physical Activities							
Coto-Montes [51]	Cross sectional	200, females 58%	Over 69 years	Set of instruments	Lipid peroxidation were associated with sarcopenia in independent older adults. The prevalence of sarcopenia was 35.3% in women and 13.1% in men. It was associated with older age, functional impairment, risk of malnutrition and use of digestive system drugs. Sarcopenia was also associated with pre-frailty and depressed mood.	Spain	NA
Potes [52]	Observational longitudinal cohort study	39, no data about gender	Over 70 years	Set of instruments	Overweight induces a progressive protein breakdown reflected as a progressive withdrawal of anabolism against the promoted catabolic state leading to muscle wasting.	Spain	NA
Coto-Montes [53]	Review				Melatonin may be beneficial in attenuating, reducing or preventing each of the symptoms that characterize sarcopenia.	NA	NA
Olaya [54]	Longitudinal study	2074, females 54.4%	Over 60 years	Set of instruments	High levels of physical activity were associated with a 51% lower risk of dying, compared with moderate physical activity. Mortality dropped by 2% for each unit increase in mobility functioning	Spain	3 years
Tomàs [55]	Longitudinal study	43, females 72.1%	Over 60 years	Battery of tests	The 6-MWT is a predictor of other functional capacities; type II diabetes influences the 6-MWT.	Portugal	3 years
Loyen [56]	Cross sectional	9509, females 55.5%	Over 20 years	Accelerometer data and socio-demographic data	23% experienced more than 10 h of sedentary time/day, and 72% did not meet the physical activity recommendations. Factors associated were older age and higher weight.	England, Portugal, Norway, Sweden	NA
Santos [57]	Cross sectional	4575, females 58.6%		Accelerometer data and socio-demographic data	Sedentary time is more than 60% of older adults’ wear time.	Portugal	NA
Pereira [58]	Cross-sectional	381	Over 75 years	Set of instruments	Institutionalization increased by 1.6% for each additional year of age. Each additional 100 MET-min/week expended on physical active decrease by 2%; Each additional meter walked in the aerobic endure test decrease by 0.9%; Each fewer unit in BMI by 24.8%.	Portugal	NA

UHS: Use of Hospital Service; UCLA: University of California, Los Angeles; WHOQOL: The World Health Organization Quality of Life; OR: Odds Ratio; CI: Confidence Interval; 6-MWT: 6-Minutes Walk Test; MET: Metabolic Equivalent of Task; NA: Not Available.

**Table 3 ijerph-15-02843-t003:** Interventions to mitigate frailty and its consequences.

Article (Ref. Number)	Name of the Intervention/Project	Aims	Target Groups	Tools/Assessment	Type of Intervention
Cano [59]	FOCUS	Contribute to the reduction of burden created by frailty by reviewing innovative practices.	Elderly and their caregivers. Stakeholders. Partners of EIP-AHA. Member States.	Quantitative approach: 2 systematic reviews; analyses of activities of EIP-AHA. Qualitative approach: metasynthesis of stakeholders’ reports.	Focus groups meetings. Structured surveys. Delphi consensus. Skype conferences. On site meetings. Virtual meetings of the network.
www.frailsafe-project.eu. [61]	FrailSafe	Quantify frailty and eventually prevent it by analysing physiological, cognitive, behavioural and social parameters in real-time.	Older person	NA	NA
Wijnhoven [63]	Prevention of Malnutrition In Senior Subject (PROMISS)	prevent protein malnutrition in community dwelling older adults in Europe.	Community-dwelling adults aged 55 years and older.	Protein screener questionnaire: it consists of questions on weight and height, and the consumption several foods selected because of their impact on protein malnutrition.	Data from 1348 older men and women (LASA study) were used to develop the questionnaire and data from 563 older man and women (HELIUS study) were used for external validation
Ramalho [64]	Community-based exercise intervention for gait and functional fitness improvement in an older population.	Evaluate at 0, 12, 24, 36 weeks if a periodic community program will have significantly results in the improvement of gait and functional fitness parameters	A total of 191 people. Inclusion criteria: ≥65 years old people; community dwelling living; understand the Portuguese language.	- SFT (Senior Fitness Tests). - Fullerton Advanced Balance Scale. - TFFS (Total Functional Fitness Score). - YPAS (Yale Physical Activity Survey).	The intervention: posture control, balance (static and dynamic), strength and agility of lower limbs and aerobic capacity for 36 weeks, twice a week, for 50 minutes each session. The control group will be composed by older people that will receive standard care.
Carnide [65]	3Fights@Edu	Promote functional capacity and independent living by empowering elderly people and their families to understand the aging process	Older adults and their families.	Massive Open Online Courses (MOOCs) providing information on ageing changes to help older adults to take decisions about Risk and actions	Three hours course (3 sessions of 1 hour) run over one week, available three times per year Provision of online materials and discussion forums
Dias [66]	H2020 i-PROGNOSIS project	Mitigate frailty by acting on Parkinson’s Disease (PD) symptoms in a personalized and gamified environment.	Patients with Parkinson’s Disease.	Targeting intelligent intervention in PD area, the Personalised Game Suite (PGS) integrates different serious games in a unified platform, namely: - ExerGames, - DietaryGames, - EmoGames, and - Handwriting/Voice Games.	Intervention platform with the integration of Serious Games to assist physical exercise, handwriting, diet improvement, and better control of emotions of PD patients.
Illario [67]	NutriLive	Promote a nutritional approach for prevention of functional decline and frailty across the whole European continent.	Inclusion criteria: ≥65 years citizens - Community dwelling - Living in assisted living facilities - inpatients at hospitals.	- Screening Assessment and Monitoring Pyramid Model (SAM-AP). - Biomarkers	An ICT platform will be set up and promoted during popular events, such as food blogger competitions on specific needs,
Arcopinto [68]	ICT-based polypharmacy management program	Give to each patient a personalized therapy that balances benefits and harms.	Older adults	Comprehensive Geriatric Assessment plus Computerized prescription support system	NA
Vuolo e Barrea [69,70]	PERSSILAA (PERsonalised ICT Supported Services for Independent Living and Active Ageing)	Develope remote service modules for: - Screening to get an overall picture of a person’s health status. - Monitoring of everyday functioning. - Training—remotely available health promotion programs.	A total of 350 over >65 years’ citizens Exclusion criteria: evident frailty, dependency in ADL, moderate to advanced dementia.	- Anthropometric measurements. - Muscle strength. - Calcaneal quantitative ultrasound scan. - PREDIMED (PREvención con DIeta MEDiterránea) questionnaire.	Health promotion and Education in community dwelling older adults
Health Directorate, Lazio Region [71]	The heat prevention plan of Lazio Region	Mitigate mortality during heat waves in frail elderly population.	The ≥65 years community dwellings with medium-high or high susceptibility to heat waves.	Susceptibility score associated to the risk of dying during heat waves, based on administrative healthcare databases or GPs clinical evaluation.	GP’s active surveillance (phone calls, home visits, other home-based treatment) during heat waves; information to patients and families during summer.

EIP-AHA: European Innovation Partnership on Active and Healthy Ageing; NA: Not Available; GP: General Practitioner.

**Table 4 ijerph-15-02843-t004:** Papers on impact evaluation of program addressing frailty in older adults.

Article (Ref. Number)	Study Design	Sample Size	Instrumental	Outcomes	Follow Up	Measure of Impact	State
Romera-Liebana [72]	RCT	A total of 347 participants Aged over 65	- Short physical performance Battery (SPPB) - Timed get up and go test (TGUGT) - Mini-Examination Cognitive of Lobo (MEC-35 Lobo); - Fried modified criteria	- Physical dimension - Neurophysiologic performance - Medication	At 3 and 18 months	Results at 3 and 18 months respectively: - SPPB score improved 1.58 and 1.36 (*p* < 0.001); - handgrip strength 2.84 and 2.49 kilogram (*p* < 0.001); - number of prescriptions decrease 1.39 and 1.09. Neurocognitive battery improved at 3 and 18 months.	Spain
Behm et al. [73]	RCT	A total of 459 persons aged over 80.	- eight frailty indicators - Mob-T Scale	- deterioration in frailty - tiredness in daily activities	2 years	Postponing the progression of frailty measured as tiredness in daily activities up to 1 year.	
Liotta [74]	Retrospective cohort study	Aged > 74 years old. A total of 6481 cases, 5724 controls	Participants to Long Live the Elderly (LLE) program	Mortality	June to September 2015	Reduction of heat-related mortality of about 13% during summer 2015	Italy
Liotta [75]	CT with historical controls	Aged ≥ 75 years old. 207 LLE program A total of 308 controls	Short Functional Geriatric Evaluation (SFGE)	Hospitalisation Mortality	Six months	Percentage of hospitalisation is 9.1% and 8.3% in the controls and in the cases respectively. LLE program reduce of about 10% the acute hospital admission rate.	Italy

RCT: Randomized Controlled Trial; CT: Controlled Trial.

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
