# Peer review of "Frailty as the Future Core Business of Public Health: Report of the Activities of the A3 Action Group of the European Innovation Partnership on Active and Healthy Ageing (EIP on AHA)"

_ijerph, 2018, doi:10.3390/ijerph15122843_

Reviewer 1 Report

This is a comprehensive summary of the work of the A3 action group that provides excellent insights on what the focus should be on frailty and public health at the community level. The authors present overwhelming evidence on the need to take a multifaceted approach. My only query is ref is made in the abstract methods section that a participatory design was included in the narrative review but very little detail is outlined in the methods section of the paper-How did this occur? What was participatory about it? Details should be included. The discussion section is very short-Would like to read more about the So what -linking in with the broader literature and current policy challenges.

A full read through proof is required for the paper-Many sentence/structures errors noted. Some below include:

Typo-Page 2 Line 55: it.-Should be it

Methods page 3 line 88-Paragraph structure needs to be fixed

Suggest that table 2, 3 and four are added as Supp file in its current state it's very large and hard to follow in the main paper.

Author Response

We thank you for all the suggestions that allowed us to improve the paper.

“This is a comprehensive summary of the work of the A3 action group that provides excellent insights on what the focus should be on frailty and public health at the community level. The authors present overwhelming evidence on the need to take a multifaceted approach. My only query is ref is made in the abstract methods section that a participatory design was included in the narrative review but very little detail is outlined in the methods section of the paper-How did this occur? What was participatory about it? Details should be included..” 

We modified the abstract method section.

“A full read through proof is required for the paper-Many sentence/structures errors noted. Some below include: Typo-Page 2 Line 55: it.-Should be it, Methods page 3 line 88-Paragraph structure needs to be fixed”

We rewrite the discussion section in order to increase the link with the current scientific debate and the policy perspectives.

“Suggest that table 2, 3 and four are added as Supp file in its current state it's very large and hard to follow in the main paper.”

We have followed your suggestion.

Reviewer 2 Report

It is unclear as to why the search strategy was limited to 2016-2018 only. 

I think the methods in the abstract could be clarified. What about this design makes it a participatory design? Who are the relevant A3 stakeholders (it appears that only research members of the A3 group were consulted when there are definite other stakeholders (members of the public, caregivers, policy makers) for whom this work impacts. It might be more clear to say that the narrative review was conducted with all A3 members vs. stakeholders. 

For Lines 74-78, perhaps a figure would help clarify the structure of the group? 

Due to the inherent nature of this paper (reporting on the actions from the workgroup), there is a risk for bias. Some of the language in the results might want to be revisited (ex. line 133: "It stands out from other tools, etc." so that there is no indication of bias in the presentation of the work. 

A reference for the validation of the frailty tools would be helpful (line 150 and 135). 

It may be helpful to provide some indication as to the public health or community level setting in which the frailty instruments presented would best be administered. For example, are these designed for primary care offices or through nursing or community settings?) 

For the section on prevalence and predictors - I'm not sure that this section contributes much new information to the literature. These predictive factors of frailty are well documented and well-understood. Perhaps more details regarding the specific greater risk (line 195/196) would be helpful. 

The discussion section could be expanded to detail some of the novel findings of the workgroup. 

There are some grammatic issues throughout and would encourage a close edit on re-submission. 

Author Response

We thank you for all the suggestions that allowed us to improve the paper.

“It is unclear as to why the search strategy was limited to 2016-2018 only.” 

We choose this time span because previous articles already described the achievements before 2016.

“I think the methods in the abstract could be clarified. What about this design makes it a participatory design? Who are the relevant A3 stakeholders (it appears that only research members of the A3 group were consulted when there are definite other stakeholders (members of the public, caregivers, policy makers) for whom this work impacts. It might be more clear to say that the narrative review was conducted with all A3 members vs. stakeholders.” 

Thank you for your suggestion, we modified the abstract according to your observation and added further explanation about the participative process.

“For Lines 74-78, perhaps a figure would help clarify the structure of the group?” 

We added a new figure to make more understandable the EIPonAHA structure.

“Due to the inherent nature of this paper (reporting on the actions from the workgroup), there is a risk for bias. Some of the language in the results might want to be revisited (ex. line 133: "It stands out from other tools, etc." so that there is no indication of bias in the presentation of the work.” 

This is true, we explained in the discussion the limitation of our paper.

“A reference for the validation of the frailty tools would be helpful (line 150 and 135)”. 

Line 135: there is no validation for this tool. Line 150: we added a reference about the RISC validation process.

“It may be helpful to provide some indication as to the public health or community level setting in which the frailty instruments presented would best be administered. For example, are these designed for primary care offices or through nursing or community settings?” 

We added a sentence about the setting of administration of the considered tools.

“For the section on prevalence and predictors - I'm not sure that this section contributes much new information to the literature. These predictive factors of frailty are well documented and well-understood. Perhaps more details regarding the specific greater risk (line 195/196) would be helpful.” 

We added some explanation about the relation between day centre activity and institutionalization. 

“The discussion section could be expanded to detail some of the novel findings of the workgroup.”

We revised the discussion in the light of your suggestion

“There are some grammatic issues throughout and would encourage a close edit on re-submission.” 

An English colleague revised the whole text.